# Evidence of Antibiotic Resistance and Virulence Factors in Environmental Isolates of *Vibrio* Species

**DOI:** 10.3390/antibiotics12061062

**Published:** 2023-06-16

**Authors:** Rajkishor Pandey, Simran Sharma, Kislay Kumar Sinha

**Affiliations:** 1Department of Biotechnology, National Institute of Pharmaceutical Education and Research, Hajipur 844102, Bihar, India; 2School of Medicine, University of Missouri, Columbia, MO 65211, USA; 3Department of Basic and Applied Sciences, National Institute of Food Technology Entrepreneurship & Management (NIFTEM), Kundli, Sonipat 131028, Haryana, India

**Keywords:** *Vibrio* species, environmental isolates, antimicrobial resistance, virulence, vibriophage

## Abstract

The outbreak of waterborne diseases such as cholera and non-cholera (vibriosis) is continuously increasing in the environment due to fecal and sewage discharge in water sources. Cholera and vibriosis are caused by different species of *Vibrio* genus which are responsible for acute diarrheal disease and soft tissue damage. Although incidences of cholera and vibriosis have been reported from the Vaishali district of Bihar, India, clinical or environmental strains have not been characterized in this region. Out of fifty environmental water samples, twelve different biochemical test results confirmed the presence of twenty *Vibrio* isolates. The isolates were found to belong to five different *Vibrio* species, namely *V. proteolyticus*, *V. campbellii*, *V. nereis*, *V. cincinnatiensis*, and *V. harveyi*. From the identified isolates, 65% and 45% isolates were found to be resistant to ampicillin and cephalexin, respectively. Additionally, two isolates were found to be resistant against six and four separately selected antibiotics. Furthermore, virulent *hlyA* and *ompW* genes were detected by PCR in two different isolates. Additionally, phage induction was also noticed in two different isolates which carry lysogenic phage in their genome. Overall, the results reported the identification of five different *Vibrio* species in environmental water samples. The isolates showed multiple antibacterial resistance, phage induction, and virulence gene profile in their genome.

## 1. Introduction

In India, many people have limited access to safe drinking water, especially in rural areas where the water sources are often untreated and contaminated with sewage/feces. Contaminated water and food are major risk factors for the spread of waterborne infectious diseases such as cholera and vibriosis. Cholera is a severe acute watery diarrheal disease caused by the O1 and O139 serogroup of *Vibrio cholerae* [1]. Apart from *V. cholerae*, many non-cholera pathogenic species of the genus *Vibrio* which cause vibriosis have been discovered in America, Europe, Asia, and other low-income countries [2,3,4]. *Vibrio* species are naturally found in freshwater, brackish water, and marine water. Generally, vibriosis infection is acquired by the ingestion of contaminated water or seafood [5]. It has been observed that environmental factors affect both eukaryotic and prokaryotic life through direct or indirect interaction. Jamie et al. mentioned in their study that warm sea surface influences the growth of pathogenic *Vibrio* species which increases the occurrence of infections in humans as well as marine animals [6]. Tropical countries such as India and other Caribbean countries are the most favorable place for the growth of toxigenic *Vibrio* species [7].

The pathogenicity of *Vibrio* species is determined by the virulent factors encoded by virulent genes. Virulence factors influence the severity of infection and drug resistance [8,9]. It has been reported that *Vibrio* species acquire external genetic material from environmental sources or other bacteria by horizontal gene transfer (HGT) [10]. The shared genetic material might be encoding virulence factors that help to enhance *Vibrio* species’ adaptability in diverse environments [11]. Virulence-associated elements such as toxin production, quorum sensing (cell to cell communication), presence of lysogenic phage, hemolysin, proteases, etc., are major pathogenic factors involved in the pathogenicity of *Vibrio* species [12,13]. Therefore, assessment of virulence factors in environmental *Vibrio* isolates provides the role of pathogenesis and helps to begin a path for treatment.

It has been reported that resistance developed by *Vibrio* species against antibiotics is associated with virulent factors [8,9]. Over time, due to the immoderate use of antibiotics in human disease, agriculture, and aquaculture, the *Vibrio* species developed antimicrobial resistance [14]. In another study, it has been characterized that the unique genetic makeup and competency of *Vibrio* species help them in adapting to adverse environmental conditions and resisting the antibacterial agent [15]. Due to the resistance developed by *Vibrio* species, they adversely affect marine, terrestrial animals, as well as human life. As reported by the Centers for Disease Control and Prevention (CDC), in mild infection of *Vibrio* species, treatment is not compulsory, but a sufficient amount of liquid should be consumed by the patient to substitute the fluid that was lost in diarrhea. In the case of mild to moderate infection of *V. cholerae*, it can be reversed by the administration of an oral rehydration solution (ORS) that helps in rehydration. For severe cholera conditions, antibacterial agents such as tetracycline, fluoroquinolones, ceftriaxone, cefotaxime, and macrolides are most effective and are used for the treatment of the disease. However, in severe vibriosis conditions, no such evidence was found that antibiotics reduce the severity of illness [16,17,18].

In earlier studies on the hotspot of cholera, most of the *Vibrio* species have been characterized in marine or brackish water. In the present study, environmental water samples in the Vaishali district of Bihar, India, have been investigated for *Vibrio* strains. Therefore, taking this into consideration, this study was performed with the objective of evaluating the presence of virulence-associated factors and antibiotic resistance in the environmental isolates of different *Vibrio* species. Additionally, it involved the characterization of phage induction in the identified isolates with the lysogenic phage-inducing agent mitomycin C (MMC).

## 2. Results

### 2.1. Isolation and Biochemical Identification of Vibrio Isolates

Out of 50 environmental water samples, 20 bacterial isolates colony (River-1, Pond-6, Stagnant water-5, Sevage-8) were picked from the TCBS plate (Figure 1A) and confirmed to be *Vibrio* positive by the 12 biochemical tests (Figure 1 and Appendix A). The biochemical test strip changed color after the addition of *Vibrio* culture, and as per the guidelines of manufacturer kit instruction (HiMedia, Mumbai, India), five different *Vibrio* species (*V. proteolyticus*, *V. campbellii*, *V. harveyi*, *V. cincinnatiensis*, *V. nereis*) were identified (Table 1).

### 2.2. Antibiotic Resistance and Susceptibility Assay

The environmental *Vibrio* isolates were checked for resistance against 14 commonly used antibiotics by disk diffusion methods (Figure 2A). Out of twenty isolates, thirteen isolates (65%) were found ampicillin resistant, and nine isolates (45%) were cephalexin resistant (Figure 2B and Appendix A). Co-trimoxazole resistance was observed in seven isolates (35%) followed by nalidixic acid in six resistant isolates (30%), as depicted in Figure 2B. Resistance against streptomycin, neomycin, cefotaxime, and furazolidone was found in one isolate each (Figure 2 and Appendix A). Antibiotics such as ciprofloxacin, gentamycin, norfloxacin, tetracycline, chloramphenicol, and polymyxin-B showed sensitivity to all isolates. Isolate VHMC-D was found to be resistant against six antibiotics namely ampicillin, cephalexin, nalidixic acid, cefotaxime, co-trimoxazole, and furazolidone. Another isolate, VHMC-B was resistant to ampicillin, cephalexin, nalidixic acid, and co-trimoxazole (Appendix A). Another seven isolates were found to be resistant to more than one antibacterial agent. Two isolates were found to be sensitive to all the antibiotics checked in the study (Appendix A).

### 2.3. Virulence Profile of Vibrio Isolates

Out of twenty *Vibrio* isolates, six isolates were chosen for the amplification of *Vibrio* genes primers as mentioned in Table 2. All the mentioned genes primers were tested for amplification with the six selected isolates. The virulence gene primer *hlyA El Tor* was found to be amplified in isolate VH-I4 (*V. proteolyticus*) at 56 °C (Figure 3A), which represents hemolysin, which is an extracellular pore-forming hemolytic toxin [19]. Another *ompW* gene primer amplified in VHMC-A (*V. campbellii*) isolate at 56 °C (Figure 3B). *ompW* is a major outer membrane protein in *Vibrio* and is involved in salt tolerance as well as in the transferring of hydrophobic small molecules [20]. However, other selected *Vibrio* gene primers were not amplified in the selected isolates.

### 2.4. Phage Induction Assessment by Mitomycin C

In the present study, it was found that after induction most of the isolates (both mitomycin C treated as well as untreated control culture) grew exponentially at comparable rates. Approximately, 2.5 h after the addition of mitomycin C (MMC+), the OD_550_ value of the induced culture of *Vibrio* isolate VH-II1 (*V. campbellii)* drastically decreased from 1.45 to 0.3 (Figure 4A). Similarly, 1.5 h after the addition of mitomycin C, the OD_550_ value of the induced culture of *Vibrio* isolate VHMC-A (*V. campbellii)* dramatically decreased from 2.05 to 0.1 (Figure 4B), whereas the control sample without mitomycin C (MMC−) continued to grow in both of the isolates (Figure 4A,B). In both isolates, typical fibers such as particles were observed in the induced culture. The significant reduction in OD at 550 nm indicated that both isolates carried lysogenic phages in their genome.

## 3. Discussion

Organic waste such as foods, sewage, fertilizers, and animal and human feces contaminate water sources and play an important role in providing the most favorable place for the growth of microbes including *Vibrio* species [27]. Further climatic changes and the enormous use of antimicrobial agents potentiate prokaryotic population and their survival in harsh environmental conditions [28].

In the present study, five different *Vibrio* species have been isolated from fresh environmental water resources. The results indicated that out of twenty isolates, eight isolates (V-Gan, VR, VM-IIA, VP-IA, VP-IB, VH-I4, VH-II4, and VHVB-I) (40%) had characteristics of *V. proteolyticus*; four isolates (VHMC-A, VM-IB, VH-I3, and VH-II1) (20%) had characteristics of *V. campbellii*; three isolates (VHMC-C, VHVB-II, and VH-II2) (15%) had characteristics of *V. cincinnatiensis*; and another three isolates (VHMC-B, VHMC-D, and VM-IA) (15%) showed *V. nereis* characteristics. There was one isolate, VH-II3,identified as *Vibrio harveyi*. A fairly mixed population of different species of *Vibrio* was observed in this region (as shown in Figure 1, Table 1 and Appendix A). In the previous study, these *Vibrio* species have been characterized as potential pathogens for humans as well as aquatic animals [1,2,5]. In an in vitro study, it was found that *V. proteolyticus* produced virulent factors cytotoxic to eukaryotic cell lines such as macrophages and HeLa cells [29]. In the year of 2017, Paek, Jayoung et al. isolated *V. cincinnatiensis* from the clinical specimen sample of a patient having symptoms of *Vibrio*-associated disease, such as watery diarrhea and soft tissue injury [30]. Another *Vibrio* species mentioned in the Harveyi Clade was considered to be the most severely pathogenic *Vibrio* cluster for aquatic animals, capable of generating more than 50 different *Vibrio* diseases [31].

Globally, antimicrobial resistance is a big challenge for the health authority. Microbes develop anti-microbial resistant genes in their genome for their protection [32]. Antimicrobial susceptibility assay of *Vibrio* isolates revealed that out of twenty isolates, thirteen (65%) and nine (45%) isolates were found to be ampicillin-resistant and cephalexin-resistant, respectively (Figure 2B and Appendix A). Those thirteen isolates were from *V. proteolyticus* [6], *V. campbellii* [2], *V. cincinnatiensis* [3], and *V. nereis* [2]. The nine isolates were from *V. proteolyticus* [5], *V. campbellii* [2], *V. cincinnatiensis* [1], and *V. nereis* [1]. Co-trimoxazole resistance was observed in seven isolates (35%) from all the isolated *Vibrio* species except *V. harveyi,* followed by nalidixic acid for which six isolates (30%) of *V. proteolyticus* and *V. nereis* were found to be resistant. Resistance against streptomycin, neomycin, cefotaxime, and furazolidone was found in one isolate each (Appendix A). The identified *Vibrio* isolates exhibited a high incidence of resistance against selected antibacterial agents that are commonly used [33]. Additionally, two isolates, VHMC-D and VHMC-B, of *V. nereis* showed resistance against six (ampicillin, cephalexin, nalidixic acid, cefotaxime, co-trimoxazole, and furazolidone) and four (ampicillin, cephalexin, nalidixic acid, and co-trimoxazole) selected antibiotics, respectively. Another seven isolates from four different *Vibrio* species except *V. harveyi* were found to be moderate to highly resistant to more than one selected antibacterial agent (Figure 2 and Appendix A). It was noted that out of twenty isolates, sixteen isolates (80%) showed resistance against beta-lactam and its derivatives. In the previous study, it was also indicated that *Vibrio* isolates had resistance against beta-lactam antibiotics and their derivatives derived from China, Italy, and the U.S. [34,35,36]. However, it was observed that two isolates, VHMC-A (*V. proteolyticus*) and VH-II3 (*V. harveyi*), were sensitive to all the selected antibiotics used in this study. Further, it was found that all the isolates were sensitive to ciprofloxacin, gentamycin, norfloxacin, tetracycline, chloramphenicol, and polymyxin-B. This finding of antibiotic resistance of the *Vibrio* isolates indicates the emergence of multidrug resistance that could be a health threat.

In general, microbes show virulence by the secretion of virulent factors which are primarily infectious to host cells. It has been observed that virulent factors of *Vibrio* species are responsible for the pathogenesis and cause diarrhea, gastroenteritis, cholera, tissue injury, and blood infections [37]. In the present study, 10 different virulent genes primers were selected for the amplification of genes in *Vibrio* isolates (Table 2). As shown in Figure 3A, the isolate (VHI4) of *V. proteolyticus* showed amplification of *hlyA* gene (hemolysin) which indicates the presence of the pore-forming toxin in the genome of the isolate. Hemolysin is a well-known pore-forming hemolytic bacterial toxin, encoded by the *hlyA gene.* This toxin is present in the genome of various pathogenic *Vibrio* species. Hemolysin is a member of the class leucocidin superfamily, it alters the host cell membrane permeability and leads to cell death by the activation of the inflammasome pathway [38]. Further, another virulent gene primer, *ompW* was found to be amplified with the genomic DNA of VHMV-A isolate (*V. campbellii*) (Figure 3B). The bacterial outer membrane protein is a major bacterial protein which is expressed on the surface of the bacterial cell membrane. It participates in the regulation of salt stress, substance transport, and osmoregulation in many *Vibrio* species [39]. OmpW expression enhances the growth of *V. cholerae* in high saline water by carnitine channel [40]. In a previous study, an increase in the OmpW expression in *V. alginolyticus* was observed in the presence of high NaCl concentration [41].

In previous studies, horizontal transfer of virulent factors has been observed in *Vibrio* through phages and other mobile genetic elements [42,43,44]. In another study, it has been mentioned that virulent genes and anti-microbial resistant genes encoded by prophage-like elements get exchanged between pathogenic and nonpathogenic *Vibrio* species [45]. In our study, interestingly, two isolates (VHMC-A and VH-II1, both were *V. campbellii*) showed a drastic decrease in the OD_550_ after 1.5 and 2.5 h of the addition of mitomycin C (MMC+), respectively, which was a typical indication of the presence of lysogenic virulent phage (Figure 4A,B). Additionally, characteristic fibers such as particles were seen in mitomycin C induced cultures (MMC+). However, without mitomycin C added, bacterial culture (MMC−) was grown constantly, and, at one point, bacterial stationary phase was established. These observations strongly suggest that both these isolates, VHMC-A and VH-II1, carry lysogenic phages in their genome. Supernatants from the induced cultures were spotted on all the isolates in order to find a suitable host that can support the entry and growth of these putative phages, but no clearing was seen which indicated that these isolates could not be infected with these phages.

## 4. Materials and Methods

### 4.1. Sample Collection and Isolation of Bacterial Colonies

A total of 50 water samples were collected in sterile 500 mL bottles from various sources such as rivers, ponds, sewage, and stagnant rainwater from different locations in the Vishal district of Bihar, India (duration: October to June). After sample collection, 50 mL of water sample was filtered through a 0.22 μm membrane filter (Millipore, Bethesda, MD, USA). The retained content was suspended in phosphate-buffered saline (PBS, pH 8.4)and then it was enriched in alkaline peptone water (APW, pH 8.4) and incubated at 37 °C for 6–8 h in an incubator shaker (ThermoFisher Scientific, Berkeley, MO, USA). Further presumptive bacterial colonies were isolated according to the method described by Mishra et al. [46], with slight modification. Briefly, the enriched subculture was streaked using a sterile inoculation loop on a thiosulfate citrate bile salts sucrose (TCBS) agar plate (Sigma Aldrich, Kenilworth, NJ, USA) and incubated at 37 °C for 18–24 h. The promising isolated smooth yellow colonies were selected and subjected to biochemical tests for further confirmation of *Vibrio* species.

### 4.2. Identification of Vibrio Isolates by Biochemical Tests Assay

The preserved bacterial colonies on agar stabs (1.5% Agar, 5 g yeast extract, 10 g tryptone, and 10 g NaCl in 1000 mL of nuclease free water, pH = 7.2) were picked up for biochemical tests. A total of 12 biochemical tests were performed for each sample. These tests include Voges–Proskauer, arginine utilization, salt tolerance, ortho-nitrophenyl-b-D-galactopyranoside (ONPG), citrate utilization, ornithine utilization, and different carbohydrate tests as mentioned in manufacturer protocol (HiMedia, Mumbai, India).

The selected colony was inoculated in a conical flask containing 5 mL alkaline peptone water and incubated at 37 °C until the inoculum turbidity became ≥0.5 OD at 600 nm. At that point, the sample was ready to be used for biochemical tests. A test strip (Hi*vibrio*^TM^ Identification kit, HiMedia, Mumbai, India) with 12 wells, each designated for a different biochemical test analysis, was used. The test strip was opened aseptically, then 50 µL of test sample was inoculated in each well of the test strip by surface inoculation method and it was incubated for 18–24 h at 37 °C. After incubation, the test strip was analyzed for different species of *Vibrio* on the basis of color changes as per the manufacturer chart description (Hivibrio^TM^ Identification kit, HiMedia, Mumbai, India).

### 4.3. Antibiotic Susceptibility Testing

Antibiotic susceptibility testing was implemented in accordance with the Kirby–Bauer disk diffusion method following the Clinical and Laboratory Standards Institute (CLSI) guidelines [47]. In brief, 150 µL of overnight *Vibrio* culture was spread aseptically on LB agar plates. *E. coli* DH5α was used as the negative control (usually *E. coli* strain K12 MG1655 is used as a negative control). The antibiotics which are most commonly used for the treatment of cholera were chosen for antibiotic susceptibility testing [15,33]. Different antibiotics discs [ampicillin (AMP, 10 μg), chloramphenicol (C, 30 μg), cefotaxime (Ce, 30 μg), ciprofloxacin (Cf, 5 μg), co-trimoxazole (Co, 25 μg), polymyxin-B (PB, 300 unit), furazolidone (FR, 100 μg), gentamycin (GEN, 10 μg), neomycin (N, 30 μg), norfloxacin (Nx, 10 μg), nalidixic acid (Na, 30 μg), tetracycline (T, 30 μg), cephalexin (CN, 30 μg), streptomycin (S, 10 μg), trimethoprim (Tr, 5 μg)] (HiMedia, Mumbai, India) were placed at certain distances on the surface of the agar plate. The plates were incubated at 37 °C for 24 h and then the diameter of the zone of inhibition of each antibiotic disc was measured.

### 4.4. Virulent Genes Amplification by Polymerase Chain Reaction (PCR)

The genomic DNA of different *Vibrio* isolates was extracted as per manufacturer protocol (QIAGEN, Germantown, MD, USA). The genomic DNA was used as a template for *Vibrio* virulent gene amplification. In the present study, specific *Vibrio* species gene primers [*ctxA*, *ctxB*, *O1rfb*, *O139rfb*, *tcpA* (*El Tor* and *Classical*), *hlyA* (*El Tor* and *Classical*), *ompW*, *ompU*, *toxR*, *zot*] were used for PCR amplification (Table 2). The PCR reaction was performed in 25 µL solution containing 10 µL PCR buffer including MgCl_2_, dNTP mix (10×) (Promega, Madison, WI, USA), 1 µL Taq DNA Polymerase (5 U/μL) (Promega, Madison, WI, USA), 1 µL each primer (10 µM), 1 µL of template DNA, and 11 µL of nuclease free water. The reaction mixture was set up on the thermal cycler (Bio-Rad, Hercules, CA, USA). The reaction conditions involved one cycle of initial denaturation at 95 °C for 5 min and 35 cycles of denaturation at 95 °C for 30 s, annealing at the annealing temperature for 1 min, extension at 72 °C for 30 s, and one cycle of final extension at 72 °C for 7 min. The PCR amplified product was run on 1% agarose gel electrophoresis for separation. The *E. coli* template DNA and their gene primer were used as a positive control for the PCR reaction.

### 4.5. Induction of Vibriophage by Mitomycin C

Induction of vibriophage by mitomycin C was performed according to the methodology mentioned by *Castillo* et al. [48], with slight modification. In brief, preserved *Vibrio* species in LB agar stab were aseptically transferred with a sterile inoculum loop to 10 mL of autoclaved LB media and incubated at 37 °C overnight at 200 rpm in an incubator shaker (ThermoFisher Scientific, Waltham, MA, USA). The overnight grown culture was sub-cultured in LB medium and incubated again at approximately 37 °C for 2 h at 200 rpm to reach the *Vibrio* culture optical density (OD_550_) of ≥0.2 at 550 nm. Phage induction was initiated by the addition of mitomycin C (at a final concentration of 0.5 µg/mL) (ThermoFisher Scientific, MA, USA) in grown culture (MMC+), except for the control (MMC−) without mitomycin C. Then, the optical density (OD_600_) of samples was measured at a time interval of 20 min for 8 h.

### 4.6. Statistics

Statistical analysis of Mitomycin-C (MMC+) induced and uninduced (MMC−) *vibrio* culture was expressed in mean ± SD (standard deviation). All statistically analyzed data were graphed using Graph Pad Prism version 8.0.2 and Microsoft 365 (2010).

## 5. Conclusions

The present study concluded that different environmental water resources of Vaishali district, Bihar, India have the presence of distinct *Vibrio* species. Most of the identified isolates showed potential drug resistance against different selected antibacterial agents which are commonly used for the treatment of cholera as well as other bacterial diseases. Additionally, the two identified isolates (VH-I4 and VHMC-A) showed virulence gene in their genome. In spite of these characterizations, two *Vibrio* isolates (VHMC-A and VH-II1) showed lysogenic phage induction as detected by the treatment of the inducing agent MMC. Furthermore, the characterization of more isolates would be helpful in identifying other virulent strains that could give more explanatory results regarding the virulent genes, phage induction, and other multidrug resistant *Vibrio* species. These findings suggest that this district might be the next hotspot of cholera and non-cholera associated diseases in the future.

## Figures and Tables

**Figure 1 antibiotics-12-01062-f001:**
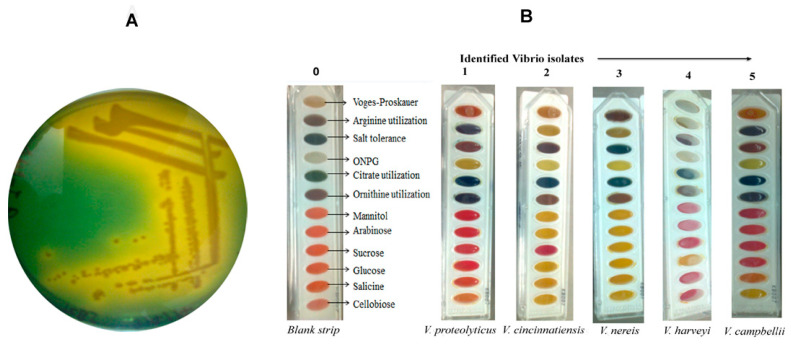
Isolation and confirmation of *Vibrio* isolates: (**A**) Image of presumed grown *Vibrio* colonies on selective TCBS agar plate. (**B**) Biochemical identification of five different *Vibrio* species. Zero indicates blank strip (without the addition of *Vibrio* culture) of 12 different biochemical tests. Numbers one, two, three, four, and five represent five different *Vibrio* species namely, *V. proteolyticus*, *V. cincinnatiensis*, *V. nereis, V. harveyi*, and *V. campbellii*, respectively.

**Figure 2 antibiotics-12-01062-f002:**
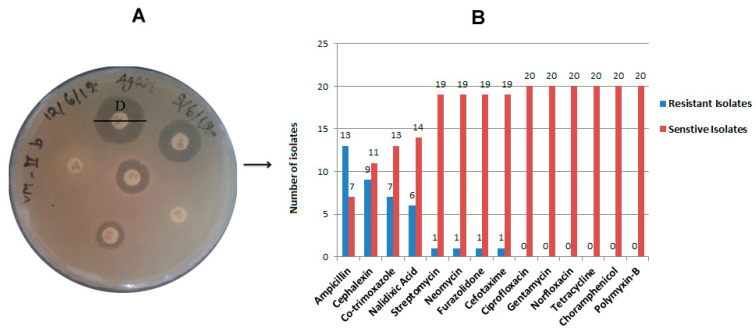
Antimicrobial susceptibility testing of identified different *Vibrio* isolates: (**A**) A representative plate of antibiotic susceptibility test assay of different *Vibrio* isolates by diffusion disk method showing the diameter of zone of inhibition. (**B**) Antimicrobial resistance and susceptibility pattern in each confirmed *Vibrio* isolates against selected antimicrobials.

**Figure 3 antibiotics-12-01062-f003:**
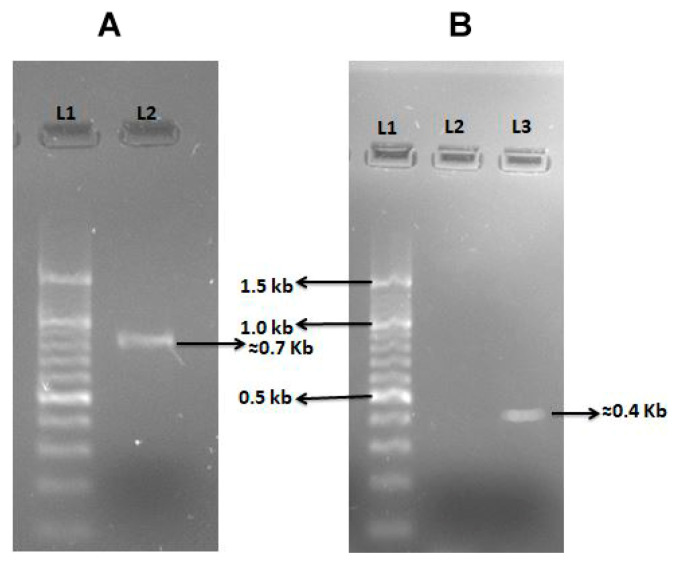
Amplification of *Vibrio* isolates’ virulent genes: (**A**) Image of 1.5% agarose gel run where Lane 1 (L1) represents 1 Kb marker and Lane 2 (L2) represents virulent gene primer hly-A (El Tor) amplification by PCR with genomic DNA of VH-I4 isolate. (**B**) View of 1.5% agarose gel run showing Lane 1 (L1): 1 kb marker, Lane 2 (L2): unamplified gene primers with selected other isolates, and Lane 3 (L3) indicating virulent gene primer *ompW* amplification by PCR with genomic DNA of isolate VHMC-A.

**Figure 4 antibiotics-12-01062-f004:**
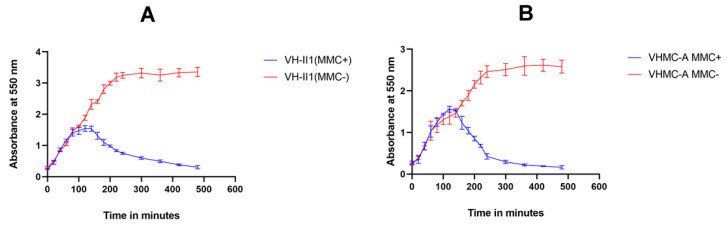
Vibriophage induction: Growth curve of vibriophage in (**A**) VH-II1 isolate and (**B**) VHMC-A isolate, after Mitomycin C induction (MMC+). The absorbance was monitored over time at 550 nm and confirmed the presence of vibriophage in VH-II1 and VHMC-A isolates.

**Table 1 antibiotics-12-01062-t001:** Different identified *Vibrio* species from different environmental water resources.

S/N	Isolate Code	Identified Species
**1**	V-Gan	*V. proteolyticus*
**2**	VR	
**3**	VH-I4	
**4**	VH-II4	
**5**	VP-IA	
**6**	VP-IB	
**7**	VM-IIA	
**8**	VHVB-I	
**9**	VH-I3	*V. campbellii*
**10**	VH-II1	
**11**	VHMC-A	
**12**	VM-IB	
**13**	VHMC-C	*V. cincinnatiensis*
**14**	VH-II2	
**15**	VHVB-II	
**16**	VHMC-B	*V. nereis*
**17**	VHMC-D	
**18**	VM-IA	
**19**	VH-II3	*V. harveyi*
**20**	VM-IIB	Not identifiable

**Table 2 antibiotics-12-01062-t002:** Primers for amplification of *Vibrio* virulent genes.

S/N	Primers	Nucleotide Sequences (5′−3′)	Amplicon Size (bp) and Annealing Temperature in °C	Reference
**1**	*ctx*A-F	CTCAGACGGGATTTGTTAGGCACG	302 (64 °C)	[21]
	*ctx*A-R	TCTATCTCTGTAGCCCCTATTACG		
**2**	ctxB-F	GATACACATAATAGAATTAAGGATG	460 (60 °C)	[22]
	ctxB-R	GGTTGCTTCTCATCATCGAACCAC		
**3**	O1 *rfb*F	GTTTCACTGAACAGATGGG	192 (57 °C)	[23]
	O1 *rfb*R	GGTCATCTGTAAGTACAAC		
**4**	O139 *rfb*F	AGCCTCTTTATTACGGGTGG	449 (57 °C)	[23]

	O139 *rfb*R	GTCAAACCCGATCGTAAAGG		
**5**	*ompW-*F	CACCAAGAAGGTGACTTTATTGTG	304 (56 °C)	[24]
	*ompW-*R	GGTTTGTCGAATTAGCTTCACC		
**6**	tcpA-F	CACGATAAGAAAACCGGTCAAGAG	451 (El Tor) (60 °C)	[25]
	tcpA-R	CGAAAGCACCTTCTTTCACACGTTG		
	tcpA-R	TTACCAAATGCAACGCCGAATG	620 (Class)	
**7**	zot-F	TCGCTTAACGATGGCGCGTTTT	947 (60 °C)	[24]
	zot-R	AACCCCGTTTCACTTCTACCCA		
**8**	hlyA*-*F	GGCAAACAGCGAAACAAATACC	481 (El Tor)	[24]
	hlyA-F	GAGCCGGCATTCATCTGAAT	738/727 (ET and Class) (56 °C)	
	hlyA-R	CTCAGCGGGCTAATACGGTTTA		
**9**	toxR-F	CCTTCGATCCCCTAAGCAATAC	779 (60 °C)	[24]
	toxR-R	AGGGTTAGCAACGATGCGTAAG		
**10**	ompU-F	ACGCTGACGGAATCAACCAAA	869 (62 °C)	[26]
	ompU-R	GCGGAAGTTTGGCTTGAAGTAG		

## Data Availability

All data generated or analyzed during this study are included in this published article in the main manuscript.

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
