# Peer review of "Evidence of Antibiotic Resistance and Virulence Factors in Environmental Isolates of Vibrio Species"

_antibiotics, 2023, doi:10.3390/antibiotics12061062_

Round 1
Reviewer 1 Report
The authors tested 50 environmental water samples and obtained 20 isolates with the Vibrio genus major characteristics. Based on HiVibrio 12 tests, 19 isolates were V. proteolyticus, V. campbellii, V. nereis, V. cincinnatiensis, and V. harveyi. The 20 isolates were tested for antibiotic resistance to 14 antibiotics, their virulence profile based in the presence of 10 virulence genes (PCR based) and presence of phages by mitomycin C. The results are presented but almost no discussion. In some cases, in the Discussion section, almost the same exact text of the results section is written.
- Change virulent to virulence every time you refer to virulence factors or genes.
Abstract
I found the first sentence a little bite difficult to read. Do you mean?:
Outbreaks related with waterborne disease such as cholera and non-cholera (vibriosis) are continuously increasing in areas with high population density due to environmental isolates.
Results
1. Line 137-138: “Out of 20 Vibrio isolates, six isolates were found for the amplification of Vibrio genes primers as mentioned in Table 2.” – it is not clear each six isolates that were positive and for each genes, please clarify.
2. Lines 139, 214, 312: correct VHI4 to VH-I4
3. Lines 140, 151, 312: VHMV-A, this isolate it is not presented in Table 1.
4. In Fig 3 B the DNA fragment pointed out has a molecular weight more similar to 400 bp than 300 bp.
5. Lines 147 and 149: Correct 1Kb to 1 kb
Discussion
1. Line 189: Since the reference it is from 2017, consider to remove “Recently” from the beginning of the sentence.
2. Line 195: There is a reference to Table 3 but there is no Table 3 in the text and in Supplementary data it is Table 2.
3. Line 223: In validation, why? Maybe you can say that the amplified fragment has around 400 bp, instead of 300 bp, in the resolution conditions tested.
4. Lines 194-203: The data in these sentences were already presented in the Results section and almost with the same text. In this section the authors should discuss if the isolates are, or not, multiresistant (considering the different antibiotics and their target in the cell), the distribution of resistance profiles by each species, for example.
5. It is not clear the genes virulence profiles of the 20 isolates. The authors say that “Antibiotic resistance developed by Vibrio species is associated with virulence factors”. Discuss the results obtained for antibiotic resistance although no virulence factors detected.
6. Altogether, the discussion of each group of data is almost inexistent. Also a global analysis of the 20 isolates is need. For example, the different isolates, globally, have the same trend in relation to all the characteristics tested (antibiotics profiles, genes tested)?
7. What can you conclude about the isolates in each of the five different species?
8. What is the probability, in each species, of (or how many) representing one same strain?
9. Since only a few isolates are mentioned, for two or three species, the others have the same characteristics for antibiotic resistance and genes presence?
Materials and Methods
1. Line 268: Change µl to µL (as in the remaining text).
2. Lines 271-275: Although it represents the same most authors use µg instead of mcg
3. Lines 288-289: Change sec to s (s is the unit for seconds in the International Unit System).
4. Line 283: Change Table 5 to Table 2
5. Line 302: Change “20 minutes for 8 hours.” to 20 min for 8 h.
6. Line 240: Correct “0.22 μM membrane filter” to 0.22 μm membrane filter
References
1. I detected three cases were a same reference has two different numbers (3=6, 13=27 and 23=35). Please correct that and check if there are any other cases.
2. Reference 23 it is not used in the text (only the reference 35).
3. Apply the italic format to the genus and species names that are written in the titles of the references.
4. Apply the italic format to the genes names that are written in the titles of the references.
5. Reference 29 – the name of the journal it is in a different format.
- Figure S1: Result of biochemical test confirmation of each Vibrio isolates – due to the poor quality of the pictures it is not relevant to include them in the document.
Globally the text will benefit if applied an English review. Some sentences are not clear maybe due to writing but I have no specific comments.
Author Response
Dear Reviewer,
I have mentioned all the suggested comments on the revised manuscript. Please see the attachment where I have put point-by-point responses to the comments.

Reviewer 2 Report
The study by Pandey et al. isolated and characterized the Vibrio strains from 50 environmental water samples from the region of Bihar, India. The authors were able to isolate 20 Vibrio isolates and the isolates were found to belong to five different Vibrio species. From the identified isolates, a reasonable high percentage were found resistant to ampicillin and cephalexin and two isolates were found to be multi-resistant (resistant to more than three antibiotics). Furthermore, two virulent factors (hlyA and ompW) were detected by PCR in two different isolates and phage induction was also noticed in two different isolates.
Overall, I found the results of this manuscript interesting but very poorly discussed. In fact, the Discussion section is merely a repetition of ideas already mentioned in the Introduction and Results sections. A proper and deeper comparison of their results with what has been found in other studies is lacking. Also, the consequences of their findings are not clear to me: is this high prevalence of Vibrio samples and their resistance profile a “wake-up call” to the proper authorities? Finally, the “Antibiotic susceptibility testing” assays were done in LB agar plates and using an E. coli DH5α as a negative control. The results seem okay for the reader to have an idea of resistances, but it should be mentioned in the Discussion that usually these tests are made in MHI and using a different E. coli strain (K12 MG1655). Finally, the authors ought to improve the bridge about climate change and their results if they want to mention that in the Discussion.
Minor points:
1 – Line 119, please re-phrase the sentence to make it clear.
2 – Figure 3. Please invert the gel in panel A and rename lanes 1 and 2. This ought to be feasible to be done in powerpoint by rotating the image.
3 – Line 174. Please rephrase the sentence since humans also have eukaryotic cells.
4 – Line 211. The verb “is” is missing from the sentence starting with “Hemolysin…”.
5 – Line 214. Please remove “primers at 56ºC annealing.” from the sentence.
6 – Lines 269-270. Please add a reference to the sentence reefing to the “antibiotics which are most commonly used for the treatment of cholera were chosen for antibiotic susceptibility testing”.
Some typos and minor errors were found. A minor revision of the English is sufficient.
Author Response
Dear Reviewer,
I have mentioned all the suggested comments point-by-point in the revised manuscript. Please see the attachment.
